# Antimicrobial Metabolites Isolated from Some Marine Bacteria Associated with *Callyspongia crassa* Sponge of the Red Sea

**DOI:** 10.3390/microorganisms13071552

**Published:** 2025-07-02

**Authors:** Amal N. Alahmari, Shahira A. Hassoubah, Bothaina A. Alaidaroos, Ahmed M. Al-Hejin, Noor M. Bataweel, Reem M. Farsi, Khloud M. Algothmi, Naheda M. Alshammari, Amal T. K. Ashour

**Affiliations:** 1Department of Biological Sciences, Faculty of Science, King Abdulaziz University, Jeddah 21589, Saudi Arabia; 2King Fahad Medical Research Center, King Abdulaziz University, Jeddah 21589, Saudi Arabia

**Keywords:** marine bacteria, antibacterial, antifungal, secondary metabolites, marine sponge, antimicrobial resistance (AMR), *Callyspongia crassa*

## Abstract

The Red Sea is rich in symbiotic microorganisms that have been identified as sources of bioactive compounds with antimicrobial, antifungal, and antioxidant properties. In this study, we aimed to explore the potential of marine sponge-associated bacteria as sources of antibacterial compounds, emphasizing their significance in combating antibiotic resistance (AMR). The crude extracts of *Micrococcus*, *Bacillus*, and *Staphylococcus saprophyticus* exhibited significant antibacterial activity, with inhibition zones measuring 12 mm and 14 mm against *Escherichia coli*, *Staphylococcus aureus*, *Candida albicans*, and other infectious strains. The DPPH assay showed that the bacterial isolates AN3 and AN6 exhibited notable antioxidant activity at a concentration of 100 mg/mL. To characterize the chemical constituents responsible for the observed bioactivity, a GC–MS analysis was performed on ethyl acetate extracts of the potent strains. The analysis identified a range of antimicrobial compounds, including straight-chain alkanes (e.g., Tetradecane), cyclic structures (e.g., Cyclopropane derivatives), and phenolic compounds, all of which are known to disrupt microbial membranes or interfere with metabolic pathways. The bioprospecting and large-scale production of these compounds are challenging. In conclusion, this study underscores the potential for marine bacteria associated with sponges from the Red Sea to be a source of bioactive compounds with therapeutic relevance.

## 1. Introduction

The rapid emergence of antibiotic-resistant bacteria poses a significant threat to global health, leading to increased mortality, prolonged hospital stays, and substantial economic burdens. The World Health Organization (WHO) identified antimicrobial resistance (AMR) as a top global public health concern, predicting that, if left unaddressed, drug-resistant infections could cause up to 10 million deaths annually by 2050 [1,2].

Marine environments, which cover over 70% of the Earth’s surface, are promising reservoirs for new bioactive compounds [3]. Sponges (phylum Porifera), which are among the oldest metazoans, host diverse microbial communities that can constitute up to 60% of their biomass [4]. These sponge-associated bacteria engage in complex symbiotic interactions with their hosts, often producing secondary metabolites with potent antimicrobial properties [5]. Many of these metabolites are believed to play a role in the host’s chemical defense mechanisms against predators, pathogens, and competing organisms [6,7].

Recent studies have demonstrated that sponge-associated bacteria are prolific sources of structurally diverse and biologically active compounds, including polyketides, peptides, alkaloids, and terpenoids [8,9]. For instance, the genus *Bacillus*, which is commonly found in marine sponges, has yielded novel antibiotics effective against multi-drug-resistant pathogens [10]. Additionally, metabolites from sponge symbionts have shown activity against fungal infections, parasitic diseases, and even cancer, underscoring their broad-spectrum therapeutic potential [11].

Advances in genomic and metabolomic technologies have significantly enhanced the discovery and characterization of these bioactive compounds. High-throughput sequencing and mass spectrometry have enabled the identification of novel biosynthetic gene clusters and metabolic pathways in sponge-associated bacteria, facilitating the exploration of their chemical diversity [12,13]. For example, recent research utilizing these techniques has uncovered new antimicrobial peptides with activity against methicillin-resistant *Staphylococcus aureus* (MRSA) and other resistant pathogens [14,15].

This article explores the potential of sponge-associated marine bacteria to be a source of antimicrobial compounds, emphasizing their significance in combating AMR. This study highlights key discoveries, the mechanisms by which these compounds exert their activity, and the challenges associated with their bioprospecting and large-scale production. We specifically aimed to explore the possibility of using sponge-associated marine bacteria as a source of antimicrobial compounds, emphasizing their significance in combating AMR. We aimed to investigate the antibacterial and antifungal potential of marine bacteria associated with sponges from the Red Sea, with a focus on isolating and characterizing bioactive metabolites with therapeutic relevance. By leveraging advanced genomic and metabolomic tools, we aim to unlock the untapped potential of these symbiotic microorganisms, offering hope in the fight against the antibiotic resistance crisis.

## 2. Materials and Methods

### 2.1. Sponge Sample Collection

On 14 November 2021, during the autumn season, prickly tube sponge samples were collected by SCUBA divers from the reef flat in the Obhur region of the Red Sea, Jeddah, Saudi Arabia (21°45′07.4″ N, 39°02′24.6″ E), at a depth of 31 m. After collection, the sponge samples were placed in sterile plastic containers filled with seawater from the surrounding area, kept in an icebox, and promptly transported to the Microbiology Laboratory Unit II at the King Fahad Medical Research Center for further analysis. The collection process was conducted by sponge experts from the Faculty of Marine Sciences at King Abdulaziz University.

### 2.2. Isolation of Marine Bacteria Associated with Sponges

Bacteria associated with the sponge samples were isolated following the method outlined by Bibi et al. (2020) [16], with slight adjustments. Three separate specimens of *Callyspongia crassa* were cut with a sterile scalpel and rinsed multiple times in sterilized artificial seawater (ASW) to remove any external contaminants or sediment. Small sections of the sponge (~1 cm^3^) were then placed in a sterile mortar and thoroughly ground with a sterile pestle to produce a homogenate.

For serial dilution, 1 mL of the sponge homogenate was added to 9 mL of ASW, vortexed for 30 s, and then serially diluted up to 10^−6^. A 100 μL sample from each dilution was spread onto four types of cultivation media designed for marine bacteria, half-strength R2A agar (½ R2A), half-strength Tryptic Soy Agar (½ TSA), half-strength nutrient agar (½ NA), and marine agar (MA), using ASW and distilled water as the preparation bases. The plates were incubated at 28 °C for 5–7 days.

The resulting bacterial colonies were examined based on their colony morphology, including the margin, texture, form, and color of the colony. To ensure purity, the colonies were subcultured on a ½ NA medium using the streaking method, and purified isolates were cultured in half-nutrient broth (½ NB). These bacterial isolates were stored in 50% glycerol at −20 °C to be used in future studies.

### 2.3. Antimicrobial Activity of Isolated Marine Bacteria

To assess the antimicrobial activity of the bacteria isolated from the sponge against bacterial and fungal pathogens, an agar well diffusion assay was conducted, following the methods of Artizzu et al. (1996) [17], with some modifications. Each bacterial isolate was cultivated in ½ NB and incubated at 27 °C with constant shaking (150 rpm) for 72 h to promote secondary metabolite production. After fermentation, the cultures were centrifuged at 6000 rpm for 15 min, and the cell-free supernatants were subjected to liquid–liquid extraction using ethyl acetate in a 1:1 (*v*/*v*) ratio with the culture supernatant.

The test microorganisms were obtained from the King Fahad Medical Research Center, Jeddah, Saudi Arabia, and included *Staphylococcus aureus* ATCC 12600, (MRSA) ATCC 33591, *Escherichia coli* ATCC 11775, a hospital-isolated *Klebsiella pneumoniae*, *Pseudomonas aeruginosa* ATCC 9027, *Aspergillus flavus*, and *Candida albicans.* Further test plates were produced for pathogenic bacteria and maintained at 37 °C for 24 h, and fungi were incubated at 30 °C for 72 h. The plates were then checked, and inhibition zones were measured and recorded. For positive controls, Amikacin (30 µg) was used against Gram-negative bacteria, and Nitrofurantoin (300 µg) was used against Gram-positive bacteria.

### 2.4. Antioxidant Properties of Sponge-Associated Bacterial Extracts with Antimicrobial Activity

The antioxidant potential of the marine bacterial extract that demonstrated antimicrobial properties was assessed using a modified DPPH (2,2-Diphenyl-1-picrylhydrazyl) free-radical-scavenging assay, following Fahmy and Abdel-Tawab, 2021, and Truong et al., 2019 [18,19]. A 0.004% DPPH solution in ethanol was prepared by dissolving 0.002 g of DPPH powder in 50 mL of ethanol. Three concentrations (100, 50, and 25 mg/mL) of the bacterial extracts were prepared via serial dilution in DMSO. For each test, 1 mL of extract was mixed with 1 mL of DPPH solution and incubated at 37 °C for 30 min. Absorbance was then measured at 517 nm using a UV–VIS spectrophotometer. Ascorbic acid (100 μg/mL) served as the positive control, and 0.004% DPPH served as the negative control. The assay was performed in duplicate, and the percentage of the DPPH scavenging effect was calculated using the following formula:$$DPPH\;scavenging\;effect\left(\%\right)=\frac{A_{0}-A_{1}}{A_{0}}\times 100$$

A_0_ is the absorbance of the negative control.

A_1_ is the absorbance of the sample.

### 2.5. Bacterial DNA Extraction and 16S rDNA Gene Analysis

The genomic DNA of each marine bacterial isolate was extracted, following a modified protocol based on Azcárate-Peril and Raya (2001) [20]. After culturing each isolate in 5 mL of ½ NB for 48 h to achieve turbidity, 1 mL of the culture was centrifuged multiple times at 10,000 rpm and 4 °C to form a pellet. The pellet was resuspended in TES buffer with lysozyme, incubated at 37 °C, and then treated with proteinase K. Following sodium acetate treatment and chloroform–isoamyl alcohol extraction, the aqueous phase was separated, mixed with isopropanol, and stored at −20 °C overnight. The DNA was pelleted via centrifugation, washed with 70% ethanol, dried, and, finally, resuspended in 30 μL of sterile water, in which it was incubated at 37 °C for 30 min. Then, the following universal primers (Macrogen, Seoul, Republic of Korea) were used for PCR amplification: 27F (5′-*AGA GTT TGA TCC TGG CTC AG*-3′) and 1492R (5′-*AAGGAGGTGATCCAGCCGCA*-3′). The PCR products of the biologically active marine strains were sent to Macrogen Inc. (Seoul, Republic of Korea) for 16S rRNA sequencing.

### 2.6. GC–MS Analysis of Bacterial Extracts with Antimicrobial Activity

The presence of secondary metabolites in the marine bacterial extracts was analyzed using gas chromatography–mass spectrometry (GC–MS). Crude extracts from the marine bacteria that demonstrated various bioactivities were sent to the Centre of Excellence in Environmental Studies at King Abdulaziz University, Jeddah, Saudi Arabia, for analysis. The GC–MS analysis was conducted using a GCMS-QP2010 PLUS instrument (Shimadzu, Kyoto, Japan), with an Agilent J&W DB-5 column (Agilent, Santa Clara, CA, USA) to optimize the separation conditions for the bacterial extracts. The temperature was programmed to rise from 110 °C to 280 °C, reaching 300 °C at the time of sample injection. Helium served as the carrier gas, maintained at a flow rate of 1 mL/min [18,21].

### 2.7. Cytotoxicity of the Bacterial Extracts with Antimicrobial Activity

The toxicity of the bacterial extracts was assessed using the brine shrimp lethality test (BSLT), following a modified procedure based on Vijayan et al. (2017) [22]. *Artemia salina* cysts were hatched in filtered sterile seawater under constant aeration and illumination at 27–30 °C for 48 h. Actively swimming nauplii were collected and acclimatized in filtered sterile seawater before use. For the assay, ten nauplii each were transferred into a sterile 60 mm Petri dish containing 5 mL of filtered seawater. The bacterial crude extracts were dissolved in dimethyl sulfoxide (DMSO; final concentration not exceeding 1%) and serially diluted to obtain three concentrations: 100, 50, and 25 mg/mL. Each concentration was added to a Petri dish to bring the final volume to 7 mL. A negative control (seawater with 1% DMSO) and a positive control (potassium dichromate, 10 μg/mL) were included. All treatments were carried out in triplicate.

The Petri dishes were incubated at 27 ± 1 °C under continuous light, and nauplii mortality was recorded at 6, 12, and 24 h. Larvae were considered dead if they exhibited no movement under a stereomicroscope, even after gentle agitation. The mortality percentage was calculated using the following formula:$$Mortality\;\%=\frac{\sum larvae\;mortality-\sum untrated\;larvae\;mortality}{\sum initail\;number\;of\;larvae}\times 100$$

Each sample was analyzed in triplicate, and the lethal concentration (LC_50_) for each extract, the concentration at which 50% of the larvae died, was determined by converting the mortality percentages to probit values.

### 2.8. Statistical Analysis

All antibacterial and antifungal experiments were performed in triplicate, and the results are expressed as means ± standard deviations (SDs). Statistical analysis was conducted using GraphPad Prism software (version 10). Descriptive statistics (mean and SD) and inferential statistics (One-Way ANOVA and Two-Way ANOVA) were applied. Differences with *p*-values < 0.05 were considered statistically significant. Two-Way ANOVA was performed, with treatment and microbial species as factors. For an in-depth comparison of each extract’s activity across strains, One-Way ANOVA was also conducted.

## 3. Results

### 3.1. Isolation of Bacteria from Sponge Samples

The sponge sample shown in Figure 1, *Callyspongia crassa*, was collected and studied for the isolation and identification of its associated bacterial symbionts. A total of 22 bacterial strains were isolated based on their colony’s morphology (shape, colony size, elevation, texture, and color). The isolated bacterial strains were designated AN1 to AN22. Eight colonies of bacteria were harvested using MA, six with ½ NA, four with ½ R2A, and another four with ½ TSA. The results indicated that ½ NA and MA were the most effective media for recovering a higher number of bacterial colonies. To obtain pure cultures, selected bacterial colonies were streaked multiple times on a ½ NA medium and incubated at 28 °C.

### 3.2. Antibacterial and Antifungal Activity of Marine Bacteria

The antimicrobial activity of the marine bacteria isolated from the *C. crassa* sponge was evaluated using the agar well diffusion method, with inhibition zones measured to assess effectiveness. Among the tested crude extracts of the isolated marine bacteria, three tested strains show significant antimicrobial activity (AN3, AN6, and AN10). AN3 exhibited the strongest activity, while AN6 showed moderate effects, and AN10 demonstrated the weakest activity. Notably, AN3 displayed the largest inhibition zone against *E. coli* (50.6 ± 0.57 mm) and the smallest inhibition zone against *C. albicans* (28 ± 1.73 mm). Similarly, AN6 showed a maximum inhibition against *E. coli* (24.6 ± 0.57 mm) and a minimum inhibition zone against *A. flavus* (18.6 ± 1.15 mm). AN10’s inhibition activity was the highest against *K. pneumoniae* (16.6 ± 1.15 mm) and the lowest against *C. albicans* (10.3 ± 0.57 mm). Figure 2 shows the inhibition zones for the bacterial extracts of AN3 against various pathogens, while Table 1 summarizes the inhibition zones of the marine isolates. Overall, 18.1% of the bacteria associated with *C. crassa* exhibited antimicrobial potential. Statistical analyses of the antimicrobial activity of the three selected bacterial extracts (AN3, AN6, and AN10) were performed using Two-Way ANOVA, as shown in Figure 3.

The antimicrobial effects of individual bacterial extracts (AN3, AN6, and AN10) against the tested microbial strains were also evaluated. One-Way ANOVA, followed by Tukey’s post hoc test, was used to identify significant differences, as shown in Figure 4, Figure 5 and Figure 6.

### 3.3. Antioxidant Activity of Crude Extracts of Active Marine Bacteria

The crude extracts of the three marine symbionts (AN3, AN6, and AN10) of the sponge *Callyspongia crassa* demonstrated notable antimicrobial activity and varying antioxidant activities. Their antioxidant potential was evaluated using the DPPH assay, in which a color change from violet to yellow and low absorbance, measured using spectrophotometry, indicated the presence of antioxidant compounds. The extract of the AN3 strain exhibited the highest antioxidant activity at a concentration of 100 mg/mL, while AN6 showed a mild effect, and AN10 displayed no significant antioxidant activity (see Table 2).

### 3.4. Identification of Active Bacteria Associated with the Sponge

According to the 16S rRNA gene sequencing, three effective bacteria associated with the sponge *Callyspongia crassa* were identified as members of the Actinobacteria and Firmicutes phyla. Strain AN3 is a member of Actinobacteria, while strains AN6 and AN10 both belong to the phylum Firmicutes. In our study, the marine bacterial isolate AN3 was identified as one strain of *Micrococcus* sp., while 16S rRNA gene sequence analysis showed that strain AN6 shared 99.37% similarity with *Bacillus amyloliquefaciens* (GenBank accession no. AB006920), suggesting a close phylogenetic relationship, and strain AN10 was identified as *Staphylococcus saprophyticus* based on 100% sequence identity of its 16S rRNA gene with the strain (GenBank accession no. NR_113956.1).

### 3.5. Cytotoxicity of the Bacterial Extracts with Antimicrobial Activity

The cytotoxicity of the bacterial extracts with antimicrobial activity was assessed, and the LC_50_ values of 83 mg/mL for the *Micrococcus* sp. extract and 50 mg/mL for *Bacillus* sp. strain AN6 were determined, as shown in Table 3. In contrast, the LC_50_ value for *Staphylococcus saprophyticus* was not determined, indicating that it exceeds 100 mg/mL. AN6 demonstrated the highest toxicity, followed by AN3, with AN10 showing the lowest toxicity (Table 3). This evaluation highlights the variance in toxicity among the bacterial extracts isolated from the sponge *Callyspongia crassa*.

### 3.6. Identification of Bioactive Secondary Metabolites Using GC–MS

Crude extracts from *Micrococcus* sp., *Bacillus* sp. strain AN6, and *Staphylococcus saprophyticus* were analyzed for bioactive secondary metabolites. The *Micrococcus* sp. isolate strongly inhibited all tested pathogenic microbes, with GC–MS analysis revealing compounds such as 1-Octadecanesulphonyl chloride (antifungal activity), two antibacterial compounds, Cyclopropane, 1,1-dichloro-2,2-dimethyl-3-(2-methylpropyl), and Tetradecane. GC–MS analysis of *Bacillus* sp. strain AN6 extract showed the presence of different chemical compounds with antimicrobial effects, including propanoic acid, ethyl ester, phenol, 3,5-bis(1,1-dimethylethyl), and Cyclopropane derivatives. Similarly, secondary metabolites from *Staphylococcus saprophyticus* (AN10) contained Tetradecane and Hexadecane (see Table 4).

## 4. Discussion

The rise of AMR poses a significant threat to global health, resulting in increases in infectious disease morbidity and mortality. In response, researchers are turning to marine sponge-associated microbes, which have shown great potential to produce novel antimicrobial compounds [8,15]. Thanks to their diverse secondary metabolites, these symbionts of sponges offer an effective defense against predators and are a promising source for new treatments against resistant pathogens. The isolation and cultivation of marine bacteria associated with the sponge *Callyspongia crassa* yielded significant insights into their biological and ecological roles. Twenty-two bacterial colonies were successfully isolated using four different culture media. Notably, the ½ NA and MA media were most effective at supporting bacterial growth, corroborating findings in [23] that nutrient-rich media such as MA favor bacterial recovery. Conversely, Ref. [16] observed that low-nutrient media like R2A can also be favorable, suggesting that nutrient availability plays a critical role in bacterial isolation. The iterative streaking and purification process ensured the isolation of pure cultures, which were verified microscopically after Gram staining.

The antimicrobial and antifungal potential of the isolated marine bacteria was evaluated using the agar well diffusion method. Three crude extracts (AN3, AN6, and AN10) demonstrated varying degrees of bioactivity, with 18.1% of isolates showing antimicrobial properties, aligning with prior bioprospecting studies that reported activity ranges of 7.5% to 34% [24,25,26]. The AN3 extract exhibited the strongest inhibition against *Escherichia coli* (50.6 ± 0.57 mm), while AN10 demonstrated the weakest activity. The bioactivity of these isolates highlights their potential for pharmaceutical applications, particularly against multidrug-resistant pathogens like MRSA. Moreover, in [26], researchers isolated 12 bacterial strains from the sponge *Xestospongia testudinaria*. Among these strains, *Bacillus subtilis* XTB-4 exhibited significant antimicrobial activity, with inhibition zones measuring 12 mm against *E. coli*, 14 mm against *S. aureus*, and 10 mm against *C. albicans*, suggesting this strain’s potential to be a source of novel antimicrobial agents. The antioxidant potential of the bacterial extracts was assessed using the DPPH assay, which revealed that AN3 and AN13 exhibited significant antioxidant activity at a concentration of 100 mg/mL. This finding aligns with the results of [27], which reported high levels of antioxidant production in *Bacillus* species isolated from marine sponges. Furthermore, *Streptomyces* sp. isolated from sponges was previously shown to secrete promising antioxidant compounds [28]. The identification of antioxidant activity in these bacterial extracts adds to their value as sources of bioactive compounds. For instance, Ref. [29] found that the co-culture of *Aspergillus* sp. CO_2_ and *Bacillus* sp. COBZ21 enhanced the production of unique metabolites, significantly boosting antimicrobial, antibiofilm, and antioxidant activities (by up to 75.25%, according to DPPH assay). GC–MS profiling revealed diverse chemical constituents, while toxicity predictions indicated no pronounced toxicity for the co-culture extract. Phylogenetic analysis based on 16S rRNA gene sequencing revealed that the bioactive isolates belong to the Actinobacteria and Firmicutes phyla, consistent with [30], which identified similar phyla among sponge-associated bacteria. The isolates AN3 and AN13 were identified as *Micrococcus* sp., AN6 as *Bacillus amyloliquefaciens* (99.37% similarity), and AN10 as *Staphylococcus saprophyticus* (100% similarity). These findings are supported by previous studies demonstrating the antimicrobial potential of *Micrococcus* and *Bacillus* species [31,32].

GC–MS analysis of the bacterial extracts identified several bioactive secondary metabolites. For instance, the extract from *Micrococcus* sp. (AN3) contained 1-Octadecanesulphonyl chloride, which is known for its antifungal activity, and Tetradecane, which has antibacterial properties. Similar findings were reported in [33,34]. The *Bacillus* sp. strain AN6 extract contained phenol, 3,5-bis(1,1-dimethylethyl), a compound with notable antimicrobial properties, as supported by [35]. The presence of Tetradecane and Hexadecane in the *Staphylococcus saprophyticus* extract (AN10) aligns with the findings of [36], further emphasizing the bioactivity of these metabolites. Selvin et al. [37] isolated the marine actinomycete *Nocardiopsis dassonvillei* MAD08 from the sponge *Dendrilla nigra*, and it showed potent antimicrobial effects against drug-resistant pathogens. Through GC–MS analysis, 11 active compounds were identified, including notable antimicrobial agents, such as 1,2-benzenedicarboxylic acid, mono(2-ethylhexyl) ester, and hexadecenoic acid. Additionally, an 87.12 kDa anticandidal protein was purified from the culture extract, demonstrating the strain’s unique ability to produce both organic and water-soluble antimicrobial substances.

The cytotoxicity assay revealed that AN6 exhibited the highest toxicity (LC_50_: 50 mg/mL), followed by AN3 (LC_50_: 83 mg/mL), while AN10 displayed the lowest toxicity (>100 mg/mL). The high toxicity of *Micrococcus* sp. (AN3) aligns with the findings of Palaniappan et al. [38], while the relatively low toxicity of AN10 contradicts the findings of Dhinakaran et al. [39]. This variation in toxicity profiles suggests a potential for the selective application of these extracts in pharmaceutical contexts.

In conclusion, this study underscores the potential of marine sponge-associated bacteria to be sources of bioactive compounds with antimicrobial, antifungal, and antioxidant properties. The isolation of strains belonging to the *Micrococcus*, *Bacillus*, and *Staphylococcus* genera and the identification of their secondary metabolites provide valuable insights into their therapeutic potential. Future research should explore the mechanisms underlying their bioactivity and evaluate their applications in drug development.

## 5. Conclusions

This study demonstrated that marine bacteria—*Micrococcus* sp., *Bacillus* sp. strain AN6, and *Staphylococcus saprophyticus*—isolated from the Red Sea sponge *Callyspongia crassa* exhibit notable antimicrobial activity. Molecular identification through 16S rRNA sequencing confirmed their taxonomic positions, while crude extracts from these strains showed potent inhibitory effects against clinically relevant bacterial and fungal pathogens. GC–MS profiling revealed the presence of diverse bioactive secondary metabolites, including propanoic acid, ethyl ester; phenol, 3,5-bis(1,1-dimethylethyl)-; and various secondary metabolites, which may contribute to their observed bioactivities. Additionally, the crude extracts exhibited moderate antioxidant properties, suggesting potential therapeutic value. The cytotoxicity assessment revealed varying LC_50_ values, with *Bacillus* sp. strain AN6 showing the highest toxicity (50 mg/mL), followed by *Micrococcus* sp. (83 mg/mL), while the *Staphylococcus saprophyticus* strain showed negligible toxicity (>100 mg/mL). Future efforts should aim to isolate and characterize the individual active compounds, assess their pharmacological profiles in vivo, and explore their underlying biosynthetic pathways using genomic and metabolomic tools. Such approaches are essential to fully harnessing the therapeutic potential of natural marine microbial products in the fight against antimicrobial resistance.

## Figures and Tables

**Figure 1 microorganisms-13-01552-f001:**
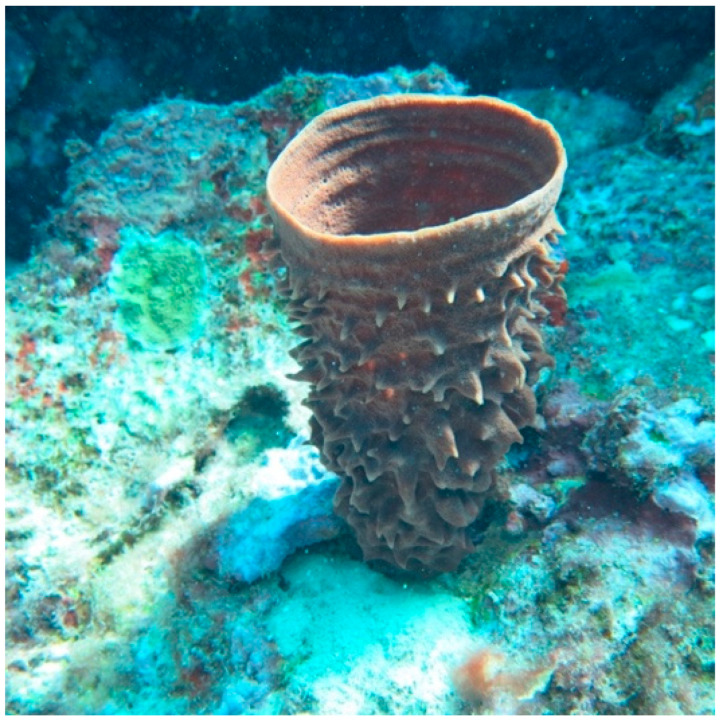
A sample of a marine sponge, *Callyspongia crassa,* collected from the Red Sea.

**Figure 2 microorganisms-13-01552-f002:**
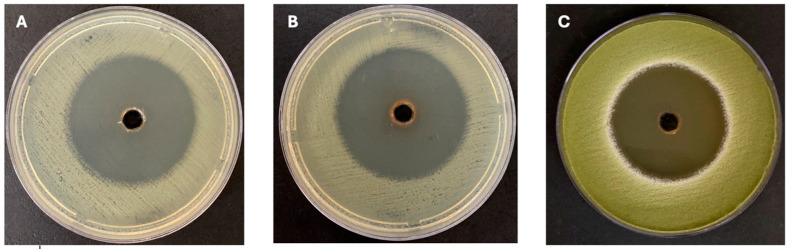
Effect of crude extracts. (**A**): AN3 extract against *E. coli*; (**B**): AN3 extract against MRSA; and (**C**): AN3 extract against *A. flavus*.

**Figure 3 microorganisms-13-01552-f003:**
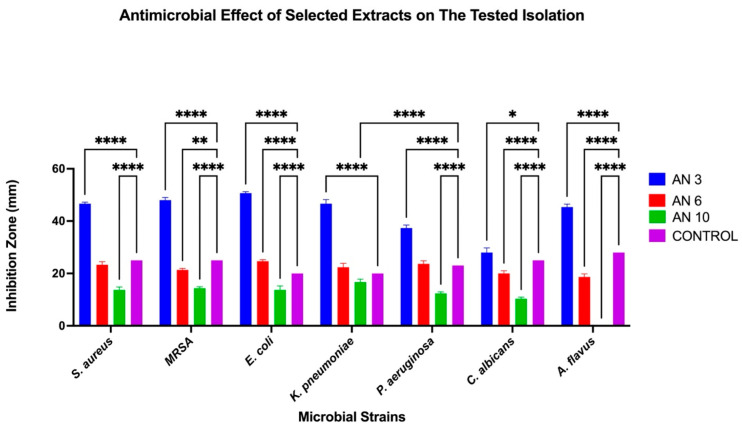
Comparative antimicrobial efficacy of three bacterial extracts and controls across tested microbial strains. Grouped bar graph showing inhibition zone diameters (mm) of *S. aureus*, MRSA, *E. coli*, *K. pneumoniae*, *P. aeruginosa*, *C. albicans*, and *A. flavus* treated with AN3 (blue), AN6 (red), and AN10 (green), along with the control (purple). Data are presented as mean ± SD (*n* = 3). Two-Way ANOVA was performed to assess the effects of treatment, microbial strain, and their interaction, followed by multiple comparisons using Tukey’s test. * *p* < 0.05; ** *p* < 0.01; **** *p* < 0.0001.

**Figure 4 microorganisms-13-01552-f004:**
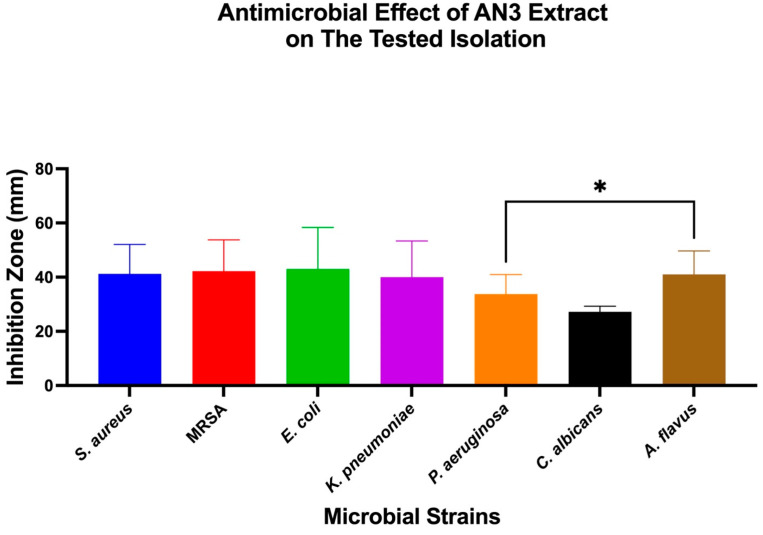
Antimicrobial activity of extract AN3 against different microbial strains. Bar chart showing mean inhibition zone diameters (mm) of *S. aureus*, MRSA, *E. coli*, *K. pneumoniae*, *P. aeruginosa*, *C. albicans*, and *A. flavus* when treated with extract AN3. Data represent mean ± SD (*n* = 3). One-Way ANOVA with Tukey’s post hoc test was used to determine statistical significance. * *p* < 0.05.

**Figure 5 microorganisms-13-01552-f005:**
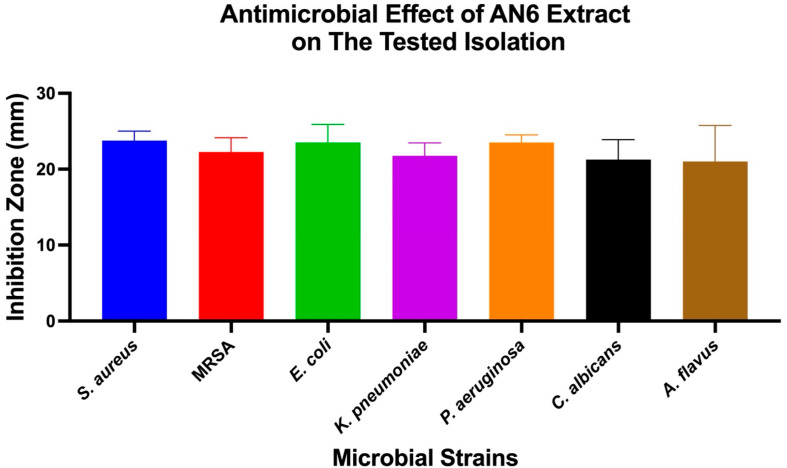
Antimicrobial activity of extract AN6 against different microbial strains. A bar chart showing mean inhibition zones (mm) for the same set of microbial strains treated with extract AN6. The results are expressed as mean ± SD (*n* = 3). Statistical analysis was conducted using One-Way ANOVA, followed by Tukey’s multiple comparison test.

**Figure 6 microorganisms-13-01552-f006:**
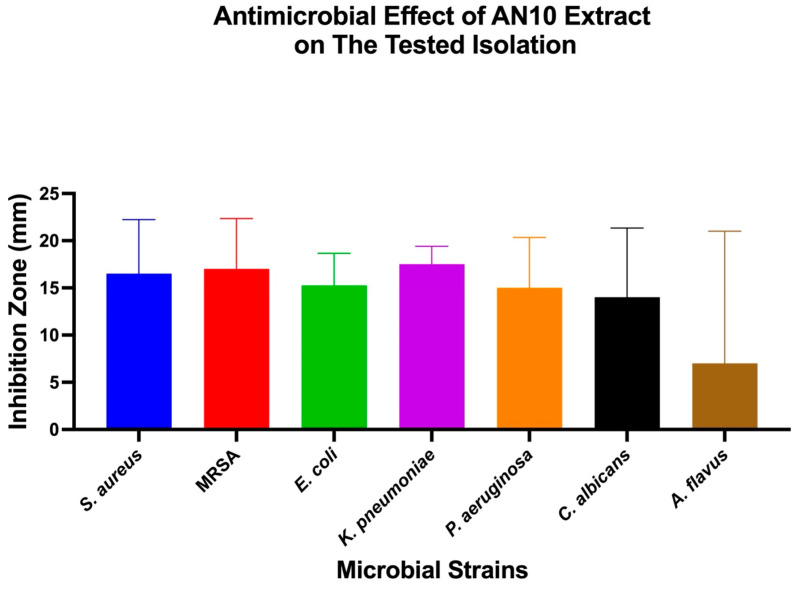
Antimicrobial activity of extract AN10 against different microbial strains. Mean inhibition zone diameters (mm) for seven tested microorganisms exposed to extract AN10. Bars represent mean ± SD (*n* = 3). One-Way ANOVA and Tukey’s post hoc analysis were used to assess differences between strains.

**Table 1 microorganisms-13-01552-t001:** Antimicrobial activity of 22 bacteria associated with *Callyspongia crassa*.

Antimicrobial Activity							
	Bacteria					Yeast	Filamentous Fungus
Sample No	*S. aureus* ^a^	MRSA ^b^	*E. coli* ^c^	*P. aeruginosa* ^d^	*K. pneumonia* ^e^	*C. albicans* ^f^	*A. flavus* ^g^
AN1	-	-	-	-	-	-	-
AN2	-	-	-	-	-	-	-
AN3	++++	++++	++++	++++	++++	+++	++++
AN4	-	-	-	-	-	-	-
AN5	-	-	-	-	-	-	-
AN6	+++	+++	+++	+++	+++	++	++
AN7	-	-	-	-	-	-	-
AN8	-	-	-	-	-	-	-
AN9	-	-	-	-	-	-	-
AN10	++	++	++	++	++	++	-
AN11	-	-	-	-	-	-	-
AN12	-	-	-	-	-	-	-
AN13	++++	++++	++++	++++	++++	+++	++++
AN14	-	-	-	-	-	-	-
AN15	-	-	-	-	-	-	-
AN16	-	-	-	-	-	-	-
AN17	+	-	-	-	-	+	-
AN18	-	-	-	-	+	-	-
AN19	-	-	++	-	+	-	-
AN20	-	-	-	-	-	-	-
AN21	-	-	-	-	-	+	-
AN22	-	-	-	-	-	-	-

^a^ Staphylococcus aureus, ^b^ Methicillin-resistant Staphylococcus aureus, ^c^ Escherichia coli, ^d^ Pseudomonas aeruginosa, ^e^ Klebsiella pneumonia, ^f^ Candida albicans, and ^g^ Aspergillus flavus. -, Negative; +, 10 mm; ++, between 11 and 20 mm; +++, between 21 and 30 mm, and ++++ between 31 and 51 mm.

**Table 2 microorganisms-13-01552-t002:** Results of DPPH assay of radical scavenging activity of active bacterial extracts.

Antioxidant Activity of Effective Bacterial Strains			
Extract Concentration	AN3	AN6	AN10
100 mg/mL	73.5%	61%	ND
50 mg/mL	70.75%	17.5%	ND
25 mg/mL	53.75%	10%	ND

**Table 3 microorganisms-13-01552-t003:** Effects of selected bacterial extracts with antimicrobial activity on the vitality of brine shrimp larvae.

Marine Bacterial Extract Concentration	AN3			AN6			AN10		
	No. of Larvae Taken	No. of Larvae Killed	LC_50_ (mg/mL)	No. of Larvae Taken	No. of Larvae Killed	LC_50_ (mg/mL)	No. of Larvae Taken	No. of Larvae Killed	LC_50_ (mg/mL)
100	10	6	83	10	10	55	10	4	>100
50	10	3	>50	10	5	50	10	2	>50
25	10	2	>25	10	4	>25	10	1	>25

**Table 4 microorganisms-13-01552-t004:** Secondary metabolites detected from crude extracts of biologically active strains.

No.	Bacteria Extract	Compound	Properties	Retention Time	Molecular Weight	Formula
1	AN3	1-Octadecanesulphonyl chloride	Antifungal	16.925	368.0	C_18_H_37_ClO_2_S
2		2,4,1-Benzoxazin-1-one, 3-trifluoromethyl-8-nitro-	-	17.419	268.1	C_14_H_11_ClF_3_NO_2_
3		Tetradecane, 2,6,10-trimethyl-	-	17.85	226.45	C_17_H_36_
4		Cyclopropane, 1,1-dichloro-2,2-dimethyl-3-(2-methylpropyl)-	Antibacterial	18.127	180.05	C_9_H_16_C_l2_
5		Tetradecane	Antibacterial	17.635	198.39	C_8_H_16_
6		Isopropyl acetate	-	6.505	102.13	C_5_H_10_O_2_
7		1-Diisopropylsilyloxy-10-undecene	-	18.35	270.5	C_16_H_36_OSi_2_
8		Phenol, 3,5-bis(1,1-dimethylethyl)-	Antimicrobial	24.766	206.32	C_14_H_22_O
9	AN6	Isopropyl acetate	-	5.403	102.13	C_5_H_10_O_2_
10		Propanoic acid, ethyl ester	Antimicrobial	6.896	102.1	C_3_H_6_O_2_
11		trans-3,4-Dimethyl-2-hexene	-	7.063	112.2	C_8_H_16_
12		2,4-Dimethyl-1-hexene	-	7.39	112.2	C_8_H_16_
13		2-Hexene, 2,5-dimethyl-	-	7.733	112.21	C_8_H_16_
14		3,4-Dimethyl-2-hexene	-	8.108	112.21	C_8_H_16_
15		Cyclopropane, 1,1-dichloro-2,2-dimethyl-3-(2-methylpropyl)-	Antibacterial	17.633	180.05	C_9_H_16_C_l2_
16		1-Diisopropylsilyloxy-10-undecene	-	18.045	270.5	C_16_H_36_OSi_2_
17		Phenol, 3,5-bis(1,1-dimethylethyl)-	Antimicrobial	6.896	206.32	C_14_H_22_O
18	AN10	Propanoic Acid, ethyl ester	Antimicrobial	6.881	102.13	C_3_H_6_O_2_
19		2-Hexene, 2,5-dimethyl-	-	7.723	112.21	C_8_H_16_
20		5-Ethyl-5-methylheptadecane	-	17.575	268.53	C_22_H_46_
21		1-Octadecanesulphonyl chloride	Antifungal	17.648	368.0	C_18_H_37_ClO_2_S
22		2,4,1-Benzoxazin-1-one, 3-trifluoromethyl-8-nitro-	-	18.064	268.1	C_14_H_11_ClF_3_NO_2_
23		Tetradecane	Antibacterial	18.229	198.39	C_14_H_30_
24		Butane, 2-(2,2-dichloro-1,3-dimethylcyclopropyl)-	-	18.473	178.0	C_9_H_16_Cl*_2_*
25		Cyclopropanecarboxylic acid, 1-(2-propenyl)-, 1,1-dimethylethyl ester	-	19.15	170.2	C_11_H_18_O_2_
26		Hexadecane	Antibacterial	19.272	226.44	C_16_H_34_
27		Tetradecane, 2,6,10-trimethyl-	-	19.725	226.45	C_17_H_36_

## Data Availability

The original contributions presented in this study are included in the article. Further inquiries can be directed to the corresponding author.

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
