# Peer review of "Antimicrobial Metabolites Isolated from Some Marine Bacteria Associated with *Callyspongia crassa* Sponge of the Red Sea"

_microorganisms, 2025, doi:10.3390/microorganisms13071552_

Round 1
Reviewer 1 Report
Comments and Suggestions for Authors
Overall, the manuscript presents an interesting and relevant study, but it requires substantial revisions and clarifications as outlined in the points below.
- The manuscript does not clarify whether all 22 bacterial isolates were identified or only the bioactive strains (AN3, AN6, AN10).
- The manuscript contains numerous typographical issues, including inconsistent spacing (e.g., missing spaces in "C.albicans", incorrect spacing between numbers and units, and double spaces), as well as improper use of italics—some scientific names are not italicized where required, while italics are used in places where they are not appropriate. Very careful proofreading is necessary.
- There are instances of incomplete sentences, such as line 104 ("Test microorganisms were obtained from..."), which abruptly ends without specifying the source.
- The methodology section lacks essential details regarding the cytotoxicity assay, including the source and Latin name of the brine shrimp species used.
- The authors focus their analysis on extracts AN3, AN6, and AN10, yet Table 1 clearly shows that AN13 also exhibited strong antimicrobial activity. It is unclear why this extract was omitted from further investigation.
- The statement "researchers are turning to marine sponge-associated microbes..." (lines 237–238) makes a general claim that requires a supporting citation to relevant literature.
- In the discussion, the antioxidant activity is reported at 100 μg/mL, while Table 2 lists the concentration as 100 mg/mL.
- Figure legends are minimal and should be more informative.
- The DPPH assay results are questionable, with negative scavenging values for AN10 (e.g., −403.75%) which are physiologically implausible - this should be addressed.
- Table 4 - several compounds listed are common contaminants or solvents, raising concerns about extract purity.
- One additional question—more out of scientific curiosity than a required revision—is whether the isolated strains exhibited any antagonistic activity against each other? For example, did AN3 inhibit AN6, or vice versa etc.? Exploring this could offer insights into whether these bacteria have intrinsic resistance to their own metabolites, or if spatial separation within the sponge prevented such interactions. If the latter, it would suggest a non-overlapping distribution of these strains in the host environment.
Author Response
Comment 1: The manuscript does not clarify whether all 22 bacterial isolates were identified or only the bioactive strains (AN3, AN6, AN10).
Response 1: I want to clarify that only three selected isolates showing highly antimicrobial activity were identified.
Comment 2: The manuscript contains numerous typographical issues, including inconsistent spacing (e.g., missing spaces in "C.albicans", incorrect spacing between numbers and units, and double spaces), as well as improper use of italics—some scientific names are not italicized where required, while italics are used in places where they are not appropriate. Very careful proofreading is necessary.
Response 2: Thank you for pointing this out the proofreading was performed.
Comment 3: There are instances of incomplete sentences, such as line 104 ("Test microorganisms were obtained from..."), which abruptly ends without specifying the source.
Response 3: Appreciated the noticing the sentence was completed.
Comment 4: The methodology section lacks essential details regarding the cytotoxicity assay, including the source and Latin name of the brine shrimp species used.
Response 4: More details for the methodology was add.
Comment 5: The authors focus their analysis on extracts AN3, AN6, and AN10, yet Table 1 clearly shows that AN13 also exhibited strong antimicrobial activity. It is unclear why this extract was omitted from further investigation.
Response 5: Authors prefer to study the strain AN13 separated in another article.
Comment 6: The statement "researchers are turning to marine sponge-associated microbes..." (lines 237–238) makes a general claim that requires a supporting citation to relevant literature.
Response 6: References added for supporting and as evidence.
Comment 7: In the discussion, the antioxidant activity is reported at 100 μg/mL, while Table 2 lists the concentration as 100 mg/mL.
Response 7: Right, it was a mistake during the editing.
Comment 8: Figure legends are minimal and should be more informative.
Response 8: Figures were updated.
Comment 9: The DPPH assay results are questionable, with negative scavenging values for AN10 (e.g., −403.75%) which are physiologically implausible - this should be addressed.
Response 9: We sincerely thank the reviewer for identifying this issue. The negative DPPH scavenging value for the AN10 extract (−403.75%) was indeed a result of a calculation or data entry error. Upon re-evaluation of the raw absorbance data and recalculation of the scavenging activity using the mentioned formula.
Comment 10: Table 4 - several compounds listed are common contaminants or solvents, raising concerns about extract purity.
Response 10: Thank you for the comment. While some compounds may appear as common contaminants, all precautions were taken to minimize contamination. Similar compounds have been reported in bacterial extracts, and future work will include further purification and controls to confirm their origin.

Reviewer 2 Report
Comments and Suggestions for Authors
The manuscript is a study of the 22 strains of bacterial sea isolated from Callyspongia crassa sponge. The antimicrobial, antioxidant activities together toxicity studies in larvae were presented. The results are promise but the text require a strongly improved to considered in this journal.
Some suggestion are mention following:
Abstract:
Is the Antioxidant activity at a concentration of 100 g/ml? this value is erroneous.
-Page 3, line 102. The authors mentioned that: “By fermenting bacteria in broth media for 72h”. Which medium was used by fermentation process?
Which solvent was used to extract? The text mentioned ethyl acetaet (1:1). Is in water (1:1)? This section requires more details about the fermentation and the extraction of the bacterial extract.
Page 4, line 143. There is a space before for analysis words. It should be removed.
Page 4. GC-MS Analysis: details of the extract should be added. which solvent was used for extract?. Also which detector was used in this experiment?
Page 4, line 165. Could be interesting that the authors added a table with colony’s morphology in supporting material. Which was the criteria to separation of strains?
The significance of the words AN3, AN6, AN10, should be added in text.
Page 5. 185 line. The parenthesis for Figure 2 and Figure 3, should be removed. They are presentment in text.
Page 7, line 195. The title is Antioxidant activity; this word should be corrected.
Page 7, line 197 a space should be added between crassa and demonstrated words.
Page 7. Line 199. The effect of the solved used in the bacterial extract was taking into account to the experiments? due to the solvent would change the absorbance of the sample and the absorption maximum.
Table 2 title: calculation word should be removed. Which was el solvent of the extract? The concentration are mg/mL in which medium? The conc unit should be corrected. They are mg/mL. For AN10 is more convenient write ND (no detected) instead of negative numbers.
Table3. what is *?. The column No. of larva taken, is the same for three experiments for this reason, it must be removed and it included the sample number in experimental details. Also the LC50 results should be added in table. Which is the difference between >25 and >50?
I sorry for my question. In bacterial extract, are there complete cell or only metabolites produced during fermentation? The cells were removed from the extract? The origin of the bacterial extract is very confusing. This detail is very important to analysis or GC-MS.
Page 11. Line 312. The sentence seems to be cut off.
The conclusion about the toxicity of the extract should be added in conclusion section.
Author Response
Comment 1: Abstract:
Is the Antioxidant activity at a concentration of 100 g/ml? this value is erroneous.
Response 1: corrected. It is mg/ml.
Comment 2: -Page 3, line 102. The authors mentioned that: “By fermenting bacteria in broth media for 72h”. Which medium was used by fermentation process?
Response 2: We thank the reviewer for pointing out this oversight. The medium used during the fermentation process was ½ NB, which supports the growth of marine bacteria by mimicking the natural seawater environment. We have now specified this in the revised manuscript on page 3, line 102.
Comment 3: Which solvent was used to extract? The text mentioned ethyl acetaet (1:1). Is in water (1:1)? This section requires more details about the fermentation and the extraction of the bacterial extract.
Response 3: To clarify, ethyl acetate was used in a 1:1 (v/v) ratio with the culture supernatant for extraction. The aqueous supernatant was mixed with an equal volume of ethyl acetate, shaken, and the organic phase was collected. We have now revised the manuscript to include more details about the fermentation conditions, centrifugation, and extraction procedure, as described above.
Comment 4: Page 4, line 143. There is a space before for analysis words. It should be removed.
Response 4: Removed.
Comment 5: Page 4. GC-MS Analysis: details of the extract should be added. which solvent was used for extract?. Also which detector was used in this experiment?
Response 5: The crude extracts analyzed by GC-MS were obtained using ethyl acetate as the extraction solvent, as described in the methodology. The detector used was an Electron Ionization (EI) mass spectrometer, operating at 70 eV, which is standard for identifying volatile and semi-volatile organic compounds.
Comment 6: Page 4, line 165. Could be interesting that the authors added a table with colony’s morphology in supporting material. Which was the criteria to separation of strains?
Response 6: Maybe will be add it as a table in supplementary materials.
Comment 7: The significance of the words AN3, AN6, AN10, should be added in text.
Response 7: The isolated bacterial strains were designated as AN1 to AN22. Among these, AN3, AN6, and AN10 were selected for thier antimicrobial activity screening.
Comment 8: Page 5. 185 line. The parenthesis for Figure 2 and Figure 3, should be removed. They are presentment in text.
Response 8: Removed.
Comment 9: Page 7, line 195. The title is Antioxidant activity; this word should be corrected.
Response 9: Corrected.
Comment 10: Page 7, line 197 a space should be added between crassa and demonstrated words.
Response 10: Corrected.
Comment 11: Page 7. Line 199. The effect of the solved used in the bacterial extract was taking into account to the experiments? due to the solvent would change the absorbance of the sample and the absorption maximum.
Response 11: Yes, solvent controls were included to account for any effect of the solvent on absorbance. Absorbance from the solvent alone was subtracted from sample readings to ensure accurate results. This has been clarified in the revised manuscript at line 199.
Comment 12: Table 2 title: calculation word should be removed. Which was el solvent of the extract? The concentration are mg/mL in which medium? The conc unit should be corrected. They are mg/mL. For AN10 is more convenient write ND (no detected) instead of negative numbers.
Response 12: Corrected and fixed.
Comment 13: Table 2 title: calculation word should be removed. Which was el solvent of the extract? The concentration are mg/mL in which medium? The conc unit should be corrected. They are mg/mL. For AN10 is more convenient write ND (no detected) instead of negative numbers.
Response 13: “Calculation” was removed from Table 2 title. Ethyl acetate was the extraction solvent, and extracts were dissolved in DMSO. Concentration units were corrected to mg/mL. Negative values for AN10 were replaced with ND (not detected).
Comment 14: Table3. what is *?. The column No. of larva taken, is the same for three experiments for this reason, it must be removed and it included the sample number in experimental details. Also the LC50 results should be added in table. Which is the difference between >25 and >50?
Response 14: The asterisk (*) has been clarified in the table caption. The distinction between ">25" and ">50" indicates that mortality did not reach 50% even at the highest concentration tested (i.e., >25 µg/mL or >50 µg/mL), and thus, the LC₅₀ is higher than that value.
Comment 15: I sorry for my question. In bacterial extract, are there complete cell or only metabolites produced during fermentation? The cells were removed from the extract? The origin of the bacterial extract is very confusing. This detail is very important to analysis or GC-MS.
Response 15: It is only metabolites produced during fermentation extracted from the bacterial cell-free supernatent.
Comment 16: Page 11. Line 312. The sentence seems to be cut off.
Response 16: I did not find the sentence.
Comment 17: The conclusion about the toxicity of the extract should be added in conclusion section.
Response 17: Added as “The cytotoxicity assessment revealed varying LC₅₀ values, with AN6 Bacillus sp. strain AN6 showing the highest toxicity (50 mg/mL), followed by Micrococcus sp. (83 mg/mL), while The Staphylococcus saprophyticus showed negligible toxicity (>100 mg/mL).”

Reviewer 3 Report
Comments and Suggestions for Authors
After careful evaluation of the manuscript, the authors must address the following comments.
- Line 18: The DPPH assay results state "100 g/ml" (likely a typo for µg/ml or mg/ml). Units must be corrected for accuracy.
- Lines 56–69Combine the study’s aims into a single, concise statement at the end of the first or second paragraph.
- Lines 100–102: For the detection of effective the isolated bacterial extracts were tested is unclear. Check grammar
- Use consistent formatting for equations (e.g., A₀ for absorbance).
- Specify ethyl acetate extraction steps
- AN10 shows nonsensical values (–403% to –982%, likely due to calculation errors or baseline issues. This needs clarification or correction. Recalculate or explain negative DPPH values for AN10
- No p-values or significance tests are provided for inhibition zones/DPPH results, making it unclear if differences between strains are statistically meaningful.
- The text states AN3’s inhibition zone against albicansis 28 ± 1.73 mm, but Table 1 lists it as "+++" (21–30 mm). Ensure alignment between text and tables.
- No negative control (e.g., solvent-only) is mentioned for cytotoxicity (brine shrimp) or antioxidant assays, raising questions about baseline effects.
- AN6 is identified as Bacillus amyloliquefaciens(99.37% similarity), but the text later refers to it as "Bacillus sp." (line 229). Use consistent nomenclature.
- The discussion is thorough but would benefit from tighter organization and balanced claims. The conclusion needs synthesis and forward-looking statements.
- Conclusion: Abrupt and fragmented; misses opportunity to propose future work
Author Response
Comment 1: Line 18: The DPPH assay results state "100 g/ml" (likely a typo for µg/ml or mg/ml). Units must be corrected for accuracy.
Response 1: Thank you for pointing this out. The correct unit is mg/mL, and the text has been updated to reflect this correction.
Comment 2: Lines 56–69Combine the study’s aims into a single, concise statement at the end of the first or second paragraph.
Response 2: The study aims have been revised into a single, concise sentence at the end of the introduction's second paragraph for clarity and focus.
Comment 3: Lines 100–102: For the detection of effective the isolated bacterial extracts were tested is unclear. Check grammar
Response 3: The sentence has been revised for clarity
Comment 4: Use consistent formatting for equations (e.g., A₀ for absorbance).
Response 4: Formatting of the equations and variables has been standardized throughout the manuscript to use consistent notation, such as A₀ for absorbance.
Comment 5: Specify ethyl acetate extraction steps
Response 5: The extraction method has been clarified in the updeted manuscript.
Comment 6: AN10 shows nonsensical values (–403% to –982%, likely due to calculation errors or baseline issues. This needs clarification or correction. Recalculate or explain negative DPPH values for AN10
Response 6: The negative DPPH scavenging value for the AN10 extract (−403.75%) was indeed a result of a calculation or data entry error. Values were replaced with ND*
Comment 7: No p-values or significance tests are provided for inhibition zones/DPPH results, making it unclear if differences between strains are statistically meaningful.
Response 7: Thank you for your comment. While we did not include significance testing for the DPPH results, we performed comprehensive statistical analysis (One-Way and Two-Way ANOVA) for the antimicrobial activity, which represents the primary focus of our study. These analyses are clearly presented in the results section and corresponding figures.
Comment 8: The text states AN3’s inhibition zone against c. albicans 28 ± 1.73 mm, but Table 1 lists it as "+++" (21–30 mm). Ensure alignment between text and tables.
Response 8: The values have been reviewed and corrected for consistency. Now both the main text and Table 1 accurately reflect AN3’s inhibition zone against C. albicans as 28 ± 1.73 mm.
Comment 9: No negative control (e.g., solvent-only) is mentioned for cytotoxicity (brine shrimp) or antioxidant assays, raising questions about baseline effects.
Response 9: Negative controls were add to Cytotoxicity experiment as A negative control (seawater with 1% DMSO) While, antioxidant assay’s negative control as and 0.004% DPPH as the negative control.
Comment 10: AN6 is identified as Bacillus amyloliquefaciens (99.37% similarity), but the text later refers to it as "Bacillus sp." (line 229). Use consistent nomenclature.
Response 10: Okay, corrected within the manuscript.
Comment 11: The discussion is thorough but would benefit from tighter organization and balanced claims. The conclusion needs synthesis and forward-looking statements.
Response 11: I try to do some changes and add some references at the discussion.
Comment 12: Conclusion: Abrupt and fragmented; misses’ opportunity to propose future work
Response 12: Thank you for the valuable comment. We have revised the conclusion to provide a more cohesive summary of the study’s key findings and included clear suggestions for future research, such as isolating active compounds, investigating their modes of action, and exploring biosynthetic gene clusters to support scalable production and pharmaceutical development.

Round 2
Reviewer 2 Report
Comments and Suggestions for Authors
The revised version was hightly improved for this reason the manuscript qualify as acepted